# How to deal with missing data in supervised deep learning?

Niels Bruun Ipsen [1]  Pierre-Alexandre Mattei [* 2]  Jes Frellsen [* 1]

## Abstract

The issue of missing data in supervised learning has been largely overlooked, especially in the deep learning community. We investigate strategies to adapt neural architectures to handle missing values. Here, we focus on regression and classification problems where the features are assumed to be missing at random. Of particular interest are schemes that allow to reuse as-is a neural discriminative architecture. One scheme involves imputing the missing values with learnable constants. We propose a second novel approach that leverages recent advances in deep generative modelling. More precisely, a deep latent variable model can be learned jointly with the discriminative model, using importance-weighted variational inference in an end-to-end way. This hybrid approach, which mimics multiple imputation, also allows to impute the data, by relying on both the discriminative and the generative model. We also discuss ways of using a pre-trained generative model to train the discriminative one. In domains where powerful deep generative models are available, the hybrid approach leads to large performance gains.

## 1. Introduction

Missing data affects data analysis across a wide range of domains and the sources of missing spans an equally wide range. Recently deep latent variable models (DLVMs, Kingma & Welling, 2013; Rezende et al., 2014) have been applied to missing data problems in an unsupervised setting (e.g. Rezende et al., 2014; Nazabal et al., 2018; Ma et al., 2018; 2019; Ivanov et al., 2019; Mattei & Frellsen,

*Equal contribution [1]Department of Applied Mathematics and Computer Science, Technical University of Denmark, Denmark [2]Université Côte d'Azur, Inria (Maasai project-team), Laboratoire J.A. Dieudonné, UMR CNRS 7351, France. Correspondence to: Niels Bruun Ipsen <nbip@dtu.dk>, Pierre-Alexandre Mattei <pierre-alexandre.mattei@inria.fr>, Jes Frellsen <jefr@dtu.dk>.

*Presented at the first Workshop on the Art of Learning with Missing Values (Artemiss) hosted by the $37^{th}$ International Conference on Machine Learning (ICML).* Copyright 2020 by the author(s).

2018; 2019; Yoon et al., 2018; Ipsen et al., 2020), while the supervised setting has not seen the same recent attention. The progress in the unsupervised setting is focused on inference and imputation in a joint model over features with missing values and can be useful as an imputation step before a discriminative model. However, this approach is not necessarily optimal in terms of minimizing a prediction error.

We propose and investigate strategies for handling missing data in the supervised learning setting, while keeping any existing discriminative neural architecture as is, by inspecting how learning curves depend on the chosen strategy. Our main contribution is a joint DLVM and discriminative model that can be trained using importance weighted variational inference.

### 1.1. Previous work

A recent attempt to handle missing data in discriminative models was done by Śmieja et al. (2018), where a Gaussian mixture model (GMM) is used as a preamble to a discriminative neural network. The GMM and discriminative model are trained jointly, and in place of any missing values the activation of the corresponding input neuron is set to the average activation over the GMM conditioned on observed values. Yi et al. (2019) tackled the issue of sparsity, and specifically large variations in sparsity, by introducing sparsity normalization. This handles issues of model output covarying with the sparsity level in the input. However, it does not address the information loss due to the missing process. Ma et al. (2018) used a permutation invariant setup to avoid imputing missing data in the input of a variational autoencoder. This approach can be readily extended to the supervised setting, using the permutation invariant setup as a modified input layer.

A review of approaches to handling missing data in (non-deep) supervised learning was given by Josse et al. (2019). Here it is shown that under some assumptions, mean imputation is consistent in the supervised setting. Le Morvan et al. (2020) investigated the case of a linear predictor on covariates with missing data, showing that in the presence of missing, the optimal predictor may not be linear and how constant imputation of each feature can be optimized with regards to the model loss.

## 2. Background and notation

Rubin (1976) introduced the framework used for describing missing processes and their relation to the observed and missing data. Le Morvan et al. (2020) and Seaman et al. (2013) have pointed out some shortcomings in the way this framework and notation are often used. We will use a notation along the lines of Le Morvan et al. (2020) here.

Assume we have a data matrix $\mathbf{X} = (\boldsymbol{x}_1, \ldots, \boldsymbol{x}_n)^{\mathsf{T}} \in \mathcal{X}^n$ that contain $n$ i.i.d. copies of the random variable $\boldsymbol{x} \in \mathcal{X}$, where $\mathcal{X} = \mathcal{X}_1 \times \cdots \times \mathcal{X}_p$ is a $p$-dimensional feature space. There is a response matrix $\mathbf{Y} = (\boldsymbol{y}_1, \ldots, \boldsymbol{y}_n) \in \mathcal{Y}^n$ that contains copies of the corresponding (possibly vector valued) response variable $\boldsymbol{y} \in \mathcal{Y}$. A missing process obscures parts of $\boldsymbol{x}$ resulting in the mask variable $\boldsymbol{s} \in \{0,1\}^p$. The positions of observed entries in the data matrix $\mathbf{X}$ are contained in a mask matrix $\mathbf{S} = (\boldsymbol{s}_1, \ldots, \boldsymbol{s}_n)^{\mathsf{T}} \in \{0,1\}^{n \times p}$ such that

$$s_{ij} = \begin{cases} 1 & \text{if } x_{ij} \text{ observed,} \\ 0 & \text{if } x_{ij} \text{ missing.} \end{cases} \tag{1}$$

Then the observed data is

$$\tilde{\mathbf{X}} = \mathbf{X} \odot \mathbf{S} + \text{na} \odot (1 - \mathbf{S}), \tag{2}$$

where $\odot$ is the Hadamard product and missing values are represented by na, defining $\text{na} \cdot x_{ij} = \text{na}$ and $\text{na} \cdot 0 = 0$. We let $\text{obs}(\boldsymbol{s})$ denote the non-zero entries of $\boldsymbol{s}$ and $\text{miss}(\boldsymbol{s})$ denote the zero-entries of $\boldsymbol{s}$, such that $\boldsymbol{x}^{\text{obs}(\boldsymbol{s})}$ are all the observed elements of $\boldsymbol{x}$ and $\boldsymbol{x}^{\text{miss}(\boldsymbol{s})}$ are all the missing elements of $\boldsymbol{x}$. For simplicity we will omit the $\boldsymbol{s}$ and write $\boldsymbol{x}^{\text{obs}}$, $\boldsymbol{x}^{\text{miss}}$ respectively, whenever the context is clear.

We distinguish between the random variables $(\boldsymbol{x}^{\text{obs}}, \boldsymbol{x}^{\text{miss}})$ and the strategies used to turn realisations of $\boldsymbol{x}^{\text{obs}}$ into complete input vectors. Specifically an imputation function $\iota$ is used $\iota(\boldsymbol{x}^{\text{obs}}) \in \mathcal{X}$, such that $\iota(\boldsymbol{x}^{\text{obs}})^{\text{obs}} = \boldsymbol{x}^{\text{obs}}$.

Finally, the goal is to minimize the prediction error by maximizing the discriminative log-likelihood

$$\ell(\phi) = \sum_{i=1}^n \log p_\phi(y_i | \boldsymbol{x}_i^{\text{obs}}, \boldsymbol{s}_i). \tag{3}$$

## 3. Training deep supervised models with missing data

We wish to compare different strategies to handling missing data in supervised deep learning, specifically a convolutional neural network for classification on images. The strategies are

- 0-imputation,

- learnable imputation,

- concatenation of information in separate channels,

- three different strategies for using a DLVM with a discriminative model, **M1**, **M2** and **M3** respectively.

We describe these approaches in the sections below.

### 3.1. Zero imputation

A simple version of constant imputation is 0-imputation, which has the intuitive appeal that the activation from the input node is zeroed out (absent). The input to the discriminative model is given by

$$\iota_0(\boldsymbol{x}^{\text{obs}}) = \boldsymbol{x} \odot \boldsymbol{s} + \mathbf{0} \odot (1 - \boldsymbol{s}). \tag{4}$$

### 3.2. Learnable imputation

In the unsupervised setting constant imputation biases marginal and joint distributions, but Josse et al. (2019) have shown that mean imputation can be consistent in the supervised setting. Furthermore, Le Morvan et al. (2020) noted that the constants can be optimized with respect to the model loss. This is the approach we take here, defining learnable parameters $\boldsymbol{\lambda} \in \mathcal{X}$ to be inserted in place of the missing data, so that

$$\iota_\lambda(\boldsymbol{x}^{\text{obs}}) = \boldsymbol{x} \odot \boldsymbol{s} + \boldsymbol{\lambda} \odot (1 - \boldsymbol{s}). \tag{5}$$

### 3.3. Concatenation in separate channels

In this work we are using a convolutional neural network for classification, so a straightforward way to merge information is to put it in separate channels in the input layer. We concatenate the following information: $\iota_0(\boldsymbol{x}^{\text{obs}})$, $\lambda$ and $\boldsymbol{s}$. In multilayer perceptrons this could instead be done by concatenating information side by side.

### 3.4. Discriminative approaches using DLVMs

Here we explore how the recent progress made in applying DLVMs to missing data problems can be utilized in the supervised learning setting. We take three different approaches:

- **M1**: We propose a joint generative and discriminative model, where a joint objective (equation (9)) ensures end-to-end training (figure 1a).

- **M2**: The model is the same as **M1**, but the generative model is first pre-trained and fixed, and then the discriminative model is trained using the joint objective.

- **M3**: The dataset imputed by a generative model (figure 1b) is given as input to a discriminative model (figure 1c).

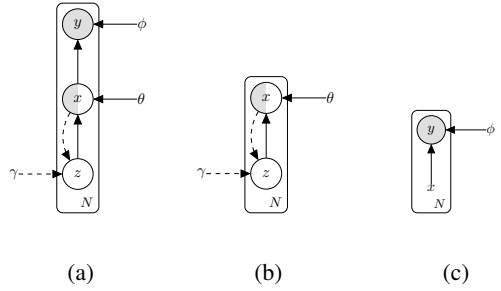

(a)        (b)        (c)

*Figure 1.* (a) Graphical model of **M1** and **M2**. For **M1** the parameters $(\theta, \phi, \gamma)$ are learnt jointly using the objective in equation (9). For **M2**, $\theta$ and $\gamma$ are found by pre-training the generative part of the model, then held fixed while learning $\phi$ using the joint loss. (b) and (c) show the approach in **M3**; the connection between the generative and discriminative model is severed, the DLVM is trained separately and used to generate a fully observed dataset as input to the discriminative model.

In all three approaches the generative parts are identical and the discriminative parts are identical. For the generative part, our choice of DLVM is the MIWAE (Mattei & Frellsen, 2019), based on importance weighted variational inference. Therefore the single imputations used to impute a full dataset are generated using self-normalized importance sampling. With the joint objective, we can utilize self-normalized importance sampling as well, but instead of weighting samples from the generative part of the model to get imputations, we are weighting the predictions.

There are subtle distinctions between imputing a fixed dataset for the discriminative model (**M3**), training the discriminative model with the joint objective (**M2**) and training both the generative and discriminative model using the joint objective (**M1**). In **M1** during training, the generative part of the model is tuned to improve the discriminative loss. In **M1** and **M2** importance weighted samples from the generative part of the model are fed to the discriminative model in place for the missing values, mimicking multiple imputation, where the class probabilities for each sample are importance weighted to give one final classification.

### 3.4.1. LOSS DERIVATION

In this section, we will derive the loss for inference in the joint model from figure 1a. The joint distribution $p(\boldsymbol{y}, \boldsymbol{x}^{\mathrm{obs}}, \boldsymbol{x}^{\mathrm{miss}}, \boldsymbol{z})$ over class labels, observed and missing covariates and latent variables can be factorized as

$$p(\boldsymbol{z})p(\boldsymbol{x}^{\mathrm{obs}}|\boldsymbol{z})p(\boldsymbol{x}^{\mathrm{miss}}|\boldsymbol{z})p(\boldsymbol{y}|\boldsymbol{x}^{\mathrm{obs}}, \boldsymbol{x}^{\mathrm{miss}}), \qquad (6)$$

where we assumed that the conditional distribution of $\boldsymbol{x}$ can be fully factorized as $p(\boldsymbol{x}|\boldsymbol{z}) = \prod_j p(x_j|\boldsymbol{z})$.

The likelihood of the observed data $p(\boldsymbol{y}, \boldsymbol{x}^{\mathrm{obs}})$ is equal to

$$\iint p(\boldsymbol{z})p(\boldsymbol{x}^{\mathrm{obs}}|\boldsymbol{z})p(\boldsymbol{x}^{\mathrm{miss}}|\boldsymbol{z})p(\boldsymbol{y}|\boldsymbol{x}^{\mathrm{obs}}, \boldsymbol{x}^{\mathrm{miss}}) \, \mathrm{d}\boldsymbol{x}^{\mathrm{miss}} \, \mathrm{d}\mathbf{z}. \qquad (7)$$

These integrals are usually analytically intractable. To approach them, we build on amortized importance-weighted variational inference (Burda et al., 2016). Indeed, the likelihood can be estimated using importance sampling

$$p(\boldsymbol{y}, \boldsymbol{x}^{\mathrm{obs}}) \approx \frac{1}{K} \sum_{i=1}^{K} \frac{p(\boldsymbol{z}_k)p(\boldsymbol{x}^{\mathrm{obs}}|\boldsymbol{z}_k)p(\boldsymbol{y}|\boldsymbol{x}^{\mathrm{obs}}, \boldsymbol{x}_k^{\mathrm{miss}})}{q(\boldsymbol{z}_k|\boldsymbol{x}^{\mathrm{obs}}, \boldsymbol{s})}, \qquad (8)$$

where $q(\boldsymbol{z}_k|\boldsymbol{x}^{\mathrm{obs}}, \boldsymbol{s})$ is the *variational distribution* (learnable proposal) and $(\boldsymbol{z}_k, \boldsymbol{x}_k^{\mathrm{miss}})_{k \in \{1, \dots, K\}}$ are i.i.d. samples from $p(\boldsymbol{x}^{\mathrm{miss}}|\boldsymbol{z})q(\boldsymbol{z}|\boldsymbol{x}^{\mathrm{obs}}, \boldsymbol{s})$. This leads to the following lower bound of the log-likelihood:

$$\mathcal{L}_K = \mathbb{E}\left[ \log\left( \frac{1}{K} \sum_{k=1}^{K} \frac{p(\boldsymbol{z}_k)p(\boldsymbol{x}^{\mathrm{obs}}|\boldsymbol{z}_k)p(\boldsymbol{y}|\boldsymbol{x}^{\mathrm{obs}}, \boldsymbol{x}_k^{\mathrm{miss}})}{q(\boldsymbol{z}_k|\boldsymbol{x}^{\mathrm{obs}}, \boldsymbol{s})} \right) \right]. \qquad (9)$$

We note that while Ipsen et al. (2020) address a very different problem, modelling data with values missing not at random, they assume the same independence structure as in figure 1a but with mask instead of label and obtain a bound with the same structure as equation (9).

**Remark.** *If a data point is fully observed, the loss is then simply*

$$\mathcal{L}_K = \log p(\boldsymbol{y}|\boldsymbol{x}) + \mathbb{E}\left[ \log\left( \frac{1}{K} \sum_{i=1}^{K} \frac{p(\boldsymbol{z}_k)p(\boldsymbol{x}|\boldsymbol{z}_k)}{q(\boldsymbol{z}_k|\boldsymbol{x})} \right) \right], \qquad (10)$$

*which is just the sum of the discriminative likelihood and the generative vanilla IWAE bound of Burda et al. (2016).*

**Remark.** *When $K = 1$, we get*

$$\mathcal{L}_1 = \mathbb{E}\left[ \log p(\boldsymbol{y}|\boldsymbol{x}^{\mathrm{obs}}, \boldsymbol{x}_1^{\mathrm{miss}}) \right] + \mathbb{E}\left[ \log\left( \frac{p(\mathbf{z}_1)p(\mathbf{x}^{\mathrm{obs}}|\mathbf{z}_1)}{q(\boldsymbol{z}_1|\boldsymbol{x}^{\mathrm{obs}}, \boldsymbol{s})} \right) \right], \qquad (11)$$

*which is the sum of a "data augmentation style" discriminative likelihood and the missing data VAE bound of Nazabal et al. (2018).*

### 3.4.2. PREDICTION

Once we have trained the model, we can perform prediction by approximating $p(\boldsymbol{y}|\boldsymbol{x}^{\mathrm{obs}}, \boldsymbol{s})$. Indeed, assuming that $p(\boldsymbol{y}|\boldsymbol{x}^{\mathrm{obs}}, \boldsymbol{s}) = p(\boldsymbol{y}|\boldsymbol{x}^{\mathrm{obs}})$, we can use self-normalised importance sampling:

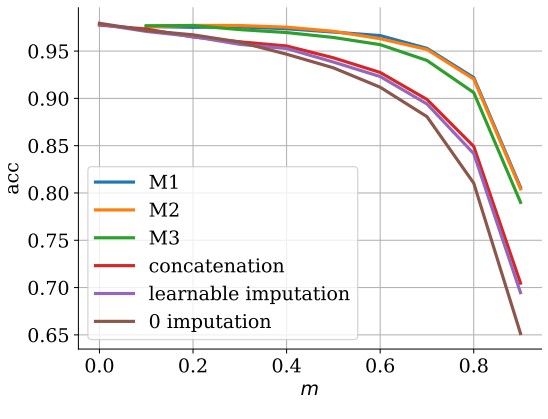

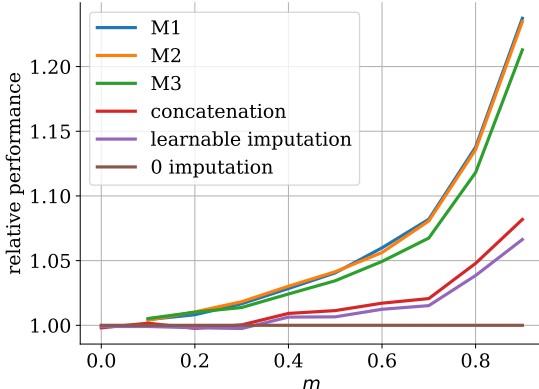

(a) Learning curves MNIST.

(b) Relative performance compared to 0-imputation.

*Figure 2.* Accuracy for different missing rates $m$. **M1** is the joint generative and discriminative model, trained jointly, **M2** is the joint model, with the generative and discriminative models trained separately and **M3** is the use of a generative model to impute the missing data to obtain a fully observed dataset, used to train a discriminative model.

$$p(\boldsymbol{y}|\boldsymbol{x}^{\mathrm{obs}}) \approx \sum_{i=1}^{K} w_k p(\boldsymbol{y}|\boldsymbol{x}^{\mathrm{obs}}, \boldsymbol{x}_k^{\mathrm{miss}}), \qquad (12)$$

where

$$w_k = \frac{r_k}{r_1 + \ldots + r_K}, \quad \text{and } r_k = \frac{p(\boldsymbol{z}_k)p(\boldsymbol{x}^{\mathrm{obs}}|\boldsymbol{z}_k)}{q(\boldsymbol{z}_k|\boldsymbol{x}^{\mathrm{obs}}, \boldsymbol{s})}, \quad (13)$$

and $(z_k, \boldsymbol{x}_k^{\mathrm{miss}})_{k \in \{1, \ldots, K\}}$ are i.i.d. samples from $p(\boldsymbol{x}^{\mathrm{miss}}|\boldsymbol{z})q(\boldsymbol{z}|\boldsymbol{x}^{\mathrm{obs}}, \boldsymbol{s})$. The prediction (seen as a probability vector) will therefore be a convex combination of the $K$ predictions obtained by imputing the data via autoencoding. Of course, $K$ should be much larger here than during training.

## 4. Experiments

We apply the different strategies for handling missing data to the dynamically binarized MNIST dataset (LeCun et al., 1998), over a range of missing rates. The discriminative model is a convolutional neural network with four hidden layers. The generative model is an MLP with two hidden layers of 200 units in the encoder and decoder, and a latent space of dimension 20. During training $K = 20$ importance samples and a batch size of 100 are used. The generative part of the models is pre-trained for 500k iterations and used as the starting point for **M1**, **M2** and **M3**.

In **M3** the pre-trained model is used immediately to generate single imputations for train, validation and test-sets, using self normalised importance sampling with 10k samples. These are then used to train the discriminative model, do early stopping and get the test-set prediction error. In

**M2** the pre-trained generative model is kept fixed while training the discriminative model using the joint loss. In **M1** the joint model is trained using the joint loss. In **M1** and **M2** predictions are done using self-normalized importance sampling on the class probabilities with 10k importance samples, cf. section 3.4.2.

Figure 2a shows that **M1** and **M2** perform best and that the performance gap increases with the missing rate. The learning curves in figure 2a obscures some of the relative performance gain, so in figure 2b the performance is shown relative to 0-imputation.

The fact that **M1** and **M2** outperform models that use single imputation indicates that accounting for uncertainty of the missing values is quite valuable.

## 5. Conclusion and future work

There are many possible approaches to deal with missing data in supervised deep learning. Our small investigations indicate that

- different ways of handling missingness may lead to quite different classification errors,

- accounting for uncertainty of the missing values can be very beneficial, even from a purely predictive perspective.

While we focused here on a simple convolutional architecture, it would be interesting to explore other kinds of architectures, from multi-layer perceptrons to recurrent/graph/group-equivariant neural nets.

## Acknowledgements

The Danish Innovation Foundation supported this work through Danish Center for Big Data Analytics driven Innovation (DABAI). JF acknowledge funding from the Independent Research Fund Denmark 9131-00082B.

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
