# OpenReview forum: "How to deal with missing data in supervised deep learning?"
_ICML.cc/2020/Workshop/Artemiss — ICML Artemiss 2020_

### Official Review · AnonReviewer1 · 2020-07-03
**Interesting contribution**

**Confidence:** 5
**Rating:** 9

**Review:**

The manuscript by Ipsen et al studies supervised learning with missing values, focusing on the specific settings of Missing At Random (MAR) values in neural networks. For this purpose, it relies on a deep latent model, learned with importance-weighted variational inference. Namely, they benchmark deep-learning approaches that train jointly a generative model of the input and a discriminative model of the output based on a simple factor model linking input data, output, and latent variables.

The manuscript is well written and an interesting addition to the literature. The main contribution over the published work is the benchmarking, which is interesting, though limited (only one dataset and one missing-values mechanism). The fact that it is didactic and cleanly formulated is very important.

I have a couple of cosmetic comments, to improve the readability of the manuscript.

First, the missing values should not be denoted as "nan", which means "not a number", and are used to track numerical errors such as overflow, underflow or division by zero. Rather, they should be written "na", which means "not available", and is the standard notation for missing values.

Second, I would find the manuscript more readable if the meaning of M1/M2/M3 were repeated a few times, for instance in figure 1 and 2.

---

### Decision · Program_Chairs · 2020-07-03

Accept